# Cetacean coronavirus spikes highlight S glycoprotein structural plasticity

Ruben J. G. Hulswit [1]*, Tatiana M. Shamorkina[2], Joline van der Lee[1], Floor Rosman[1], Lisanne S. Wetzels[1], Frank J. M. van Kuppeveld[1], Joost Snijder[2], Berend Jan Bosch[1]*, Daniel L. Hurdiss[1]*

1 Virology Section, Infectious Diseases and Immunology Division, Department of Biomolecular Health Sciences, Faculty of Veterinary Medicine, Utrecht University, Utrecht, the Netherlands, 2 Biomolecular Mass Spectrometry and Proteomics, Bijvoet Center for Biomolecular Research, Department of Chemistry, Faculty of Science, Utrecht University, Utrecht, the Netherlands

* r.j.g.hulswit@uu.nl (RJGH); b.j.bosch@uu.nl (BJB); d.l.hurdiss@uu.nl (DLH)

## Abstract

Coronaviruses (CoVs) exhibit a remarkable ability for spill-over infections into naive host populations. While much research has focused on the spike (S) glycoproteins of zoonotic alpha- and betacoronaviruses, the S proteins of gamma- and deltacoronaviruses, which predominantly infect avian hosts, remain poorly understood. Here, we present high-resolution cryo-EM structures of S proteins from two distinct gammacoronaviruses (75.7% sequence identity) that atypically infect marine mammals and belong to the Gammacoronavirus delphinapteri species. The cryo-EM reconstructions reveal that the spikes exhibit a unique quaternary architecture that distinguishes them from other coronaviruses. The S protein features a previously unidentified, tripodal quaternary assembly of the S1 subunit, in which S1$^B$ domains are presented in an upright position while their putative receptor binding sites are shielded by extended loops from the S1$^A$ domain of the same protomers. Additionally, the CeCoV spike proteins have evolved an additional and unique ~200 residue N-terminal domain (S1$^0$). S1$^0$ lacks homology to known protein sequences but displays structural similarity to members of the cupin protein superfamily. This represents a remarkable case of coronaviral exaptation of a host protein integrated into the S glycoprotein. Moreover, glycoproteomic analyses reveal that CeCoV S proteins are extensively N-glycosylated (>100 N-glycans per trimer), with a notable abundance of high-mannose glycans on S1$^0$ and O-glycosylation sites within a mucin-like loop at the trimer apex, all contributing to a dense glycan shield and potentially masking immunogenic epitopes. These findings demonstrate the structural diversity and adaptability of CoV S proteins, including alternative quaternary assemblies, additional domains, and diverse glycosylation strategies, offering new insights into the evolutionary mechanisms that enable coronaviruses to expand their host range and establish infections in novel species.

**Data availability statement:** o Atomic coordinates are deposited in the Protein Data Bank under accession codes 9R1Q (BwCoV S) and 9R1R (BdCoV S). The corresponding EM density maps (final unsharpened, sharpened, local resolution filtered, half maps, N-linked glycan difference map, and mask) have been deposited to the Electron Microscopy Data Bank under the accessions EMD-53512 (BwCoV S) and EMD-53513 (BdCoV S). The raw LC-MS/MS files and glycoproteomics analyses have been deposited to the ProteomeXchange Consortium via the PRIDE partner repository with the dataset identifier PXD063489. All reagents and relevant data are available from the authors upon request (please contact the corresponding authors: r.j.g.hulswit@uu.nl or d.l.hurdiss@uu.nl). Figshare Data 1 and 2 are accessible through public repository FigShare (https://doi.org/10.6084/m9.figshare.30156202.v1).

**Funding:** o This study was supported by the Dutch Research Council (NWO) NEMI grant 184.034.014 (to DLH), by the Dutch Research Council (NWO) XS-grant OCENW.XS22.3.110 (to RJGH), by the Innovative Medicines Initiative CARE project No. 101005077 (to BJB and FJMvK), by the Royal Netherlands Academy of Arts and Sciences (KNAW) Beijerinck Premium 2023 (to DLH), and by the Dutch Research Council (NWO) Gravitation grant 024.002.009 (to TMS and JS). The funders had no role in study design, data collection and analysis, decision to publish, or preparation of the manuscript.

## Author summary

Coronaviruses are well known for their ability to jump between species, a process driven mainly by the spike protein on their surface. In this study, we examined spike proteins from coronaviruses found in marine mammals (whales and dolphins) to better understand how these viruses evolve and interact with their hosts. Using structural biology and mass spectrometry, we discovered several unusual features of the cetacean spike proteins. These spikes contain an extra domain within their receptor-binding region that resembles cupin proteins, assemble into trimers in a novel way that hides the receptor-binding domains, and show clusters of sugars (O-linked glycans) near the sites thought to interact with host receptors. These distinctive features suggest that cetacean coronaviruses may use a different entry mechanism, in which the extra domain acts as a switch to reveal the hidden receptor-binding regions. Such a mechanism could affect which species these viruses can infect, how easily they cross species barriers, and how they evade the immune system. Our findings emphasize the structural diversity and modularity of coronavirus spike proteins, highlight the unique adaptations of coronaviruses in marine mammals, and provide insight into their potential to give rise to new diseases in other species.

## Introduction

Coronaviruses (CoVs) are a diverse family of enveloped, positive-sense RNA viruses known for their ability to infect a wide range of host species, including mammals and birds. Over the past two decades, the emergence of zoonotic coronaviruses such as SARS-CoV, MERS-CoV and SARS-CoV-2 has demonstrated their potential to spill over into new host populations, often with significant public health consequences. A key mediator of this adaptability is the spike (S) glycoprotein, a class I fusion protein that assembles into trimers and resides on the surface of CoV virions. These protein assemblies facilitate host cell entry by binding to cellular receptors and drive membrane fusion by refolding to a lower-energy conformation. As such, the S protein is the main determinant of CoV host, tissue and cell tropism and forms the key target of neutralizing antibodies for the host immune response. Accordingly, the structure and function of the S glycoprotein have been extensively studied over the past decades. Where most research has focused on the S proteins of viruses from the *Alpha-* and *Betacoronavirus* genera, those of *Gamma-* and *Deltacoronavirus* taxa are comparatively understudied.

Gammacoronaviruses predominantly infect avian species [1], with Infectious Bronchitis Virus (IBV) posing a major health threat for poultry. Members of the *Gammacoronavirus delphinapteri* species, within the subgenus *Cegacovirus* [2], present an exception due to their exclusive detection in cetaceans. These viruses - collectively referred to here as CeCoVs - have been detected in faecal samples from bottlenose dolphins (*Tursiops truncatus*) [3,4], in the liver of a beluga whale (*Delphinapterus*

*leucas*) with generalized pulmonary disease and terminal acute liver failure [5] and in the heart tissue of a striped dolphin (*Stenella coeruleoalba*) [6]. While CeCoVs have been detected in geographically distinct locations and host species, little is known about their molecular biology [7,8], host range, pathology and epidemiology.

Coronavirus spikes are typically composed of a membrane-anchored trimeric S2 fusion stalk crowned by three receptor-binding S1 subunits encompassing a multi-domain architecture that includes the canonical domains S1$^A$ through S1$^D$. The CeCoV spike protein is among the largest known coronavirus spike proteins at a length of nearly 1,500 amino acids. Two CeCoV S types have been identified, corresponding to the beluga whale and bottlenose dolphin host species [3,5]. The two types display ~75% amino acid sequence identity, with variations chiefly in the S1 subunit. CeCoV S is uniquely positioned within the CoV spike sequence space, displaying an approximately equal evolutionary distance positioned from gamma- and alphacoronavirus spikes, with only 35% protein sequence identity to its closest relatives. Intriguingly, the ~500 most N-terminal residues lack homology to known protein sequences, with sequence homology exhibited from domain S1$^B$ onwards. With currently known coronavirus S1$^A$ domains comprising ~200–300 amino acids, CeCoV spike may contain an enlarged S1$^A$ domain and/or carry an extra N-terminal domain. Such N-terminal spike extensions are known to occur among alphacoronaviruses, in which this additional domain (S1$^0$) has arisen through an S1$^A$ gene duplication event [9].

Here, we present high-resolution cryo-electron microscopy (cryo-EM) structures of the two distinct spike glycoproteins of the species *Gammacoronavirus delphinapteri*. The structures reveal an unprecedented quaternary organization, characterized by upright oriented S1$^B$ domains shielded by S1$^A$ protrusions, and a unique ~200-residue additional N-terminal domain of cellular origin. Glycoproteomic analyses reveal extensive N-glycosylation, and O-glycosylation within a mucin-like loop at the trimer apex. Our findings highlight the structural plasticity of the coronavirus S protein that drives the cross-species transmission potential of this group of medically relevant pathogens.

## Results

To explore the architecture and evolution of the cetacean coronavirus S protein, we determined the ectodomain structure of two distinct CeCoV S types (75.7% amino acid sequence identity) from the beluga whale coronavirus (BwCoV) [5] and the bottlenose dolphin coronavirus (BdCoV) [3] by cryo-EM (Fig 1a). S ectodomains of both viruses were expressed and purified and exhibited the expected molecular weight of ca. 240 kDa on reducing SDS-PAGE gels (Fig A in S1 File). The pre-fusion structures of the BwCoV and BdCoV S ectodomains are resolved at a global resolution of 2.27 and 2.65 Å, respectively (Table A and Figs B and C in S1 File for a detailed overview of the cryo-EM data processing of both maps). Both trimeric assemblies adopt a conical shape of approximately 175 Å in height and 150 Å in width (Fig 1b and 1c).

### Trimeric CeCoV spike structures display a tripodal S1 architecture centred around an upwardly tilted S1$^B$ domain

The multi-domain S1 subunit of the CeCoV spike protein comprises the canonical S1 domains (S1$^A$-S1$^D$) and an additional, atypical N-terminal domain (S1$^0$) that is not found in related gammacoronaviruses (Fig 2a). The S1$^0$ domain is juxtaposed with the S1$^A$ domain of the same protomer, with an interface that is stabilized by hydrogen bonds and Van der Waals interactions (interface area S1$^0$:S1$^A$ BwCoV: 1,300 Å$^2$, BdCoV: 1,400 Å$^2$) (Figs 2a and 3c). Structural superposition indicates that CeCoV S1$^0$ overlaps with the space occupied by S1$^A$ in IBV spike [10], seemingly contributing to the relative inward movement of S1$^A$ towards S1$^B$. S1$^A$ now appears to function as a cap of the topologically conserved receptor-binding motif (RBM) on S1$^B$, similar to what is observed for alphacoronavirus spikes [9,11]. This interaction is facilitated by an extended loop of S1$^A$ that forms the apex of the spike protein near the three-fold symmetry axis (Fig 2b). Curiously, the CeCoV S1$^B$ domain is tilted upward in a manner reminiscent of, yet different from, that of the S1$^B$ 'up'-state observed in S proteins of emerging zoonotic betacoronaviruses [12,13] (Fig 2a). Markedly, CeCoV S1$^B$ is observed in a single state and is rotated ~25° outward over the S1$^B$-S1$^C$ hinge relative to IBV S1$^B$ (Fig 2b). The S1$^A$ domains are intercalated with two neighbouring S1$^B$ domains, consequently diminishing the exposed S1$^B$ surface area by ~900–1,000 Å$^2$ (Fig 2b). Thus,

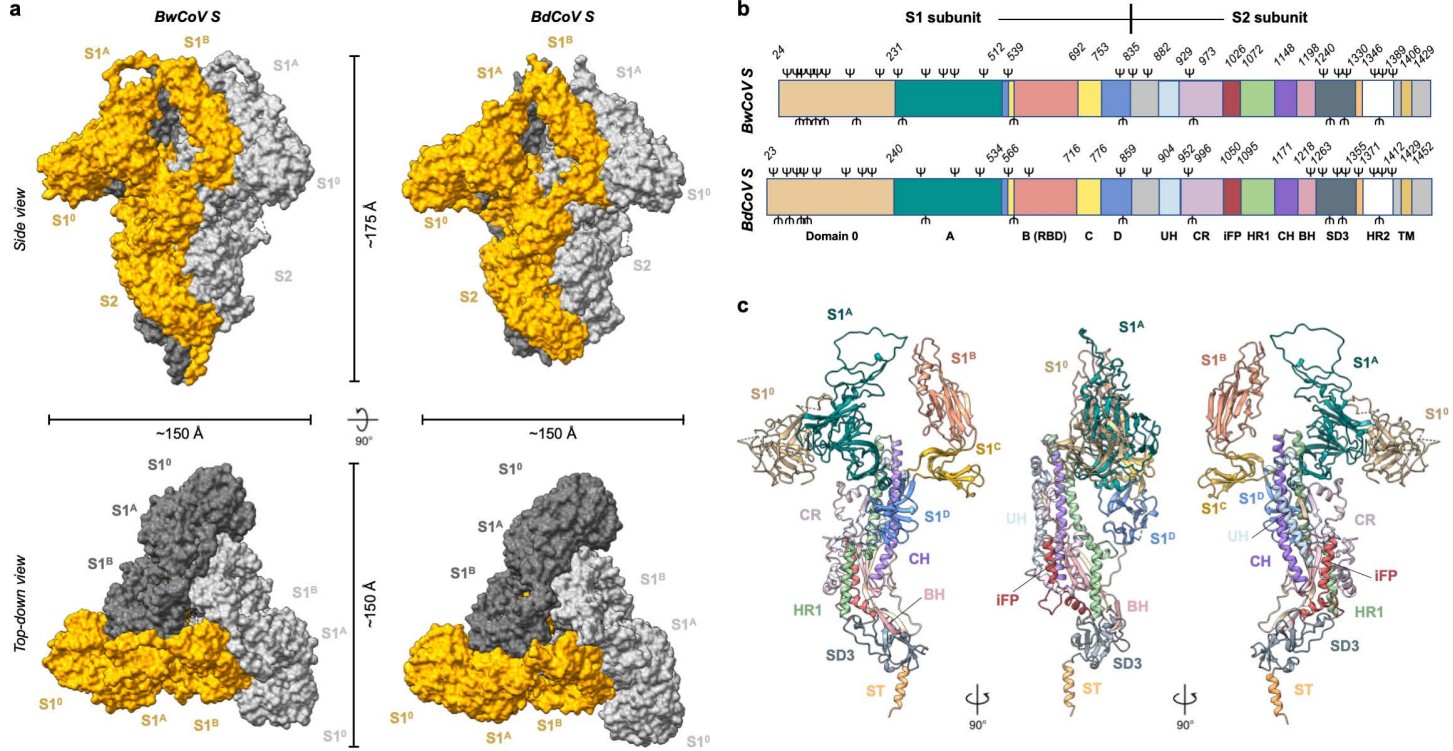

**Fig 1. Structure determination of BwCoV SED at 2.27 Å and BdCoV SED at 2.65 Å by single particle analysis. (A)** Surface presentation of atomic models of the trimeric BwCoV and BdCoV spike protein ectodomains shown as side view (upper panel) and top-down view (lower panel). Three protomers of the spike assemblies are shown in gold, black and grey. Domains and dimensions (in Å) are indicated. **(B)** Schematic linear representation of the two CeCoV S protein types studied (*upper*, BwCoV S (1,472 residues, 37 predicted N-glycosylation sequons); *lower*, BdCoV S (1,495 residues, 35 predicted N-glycosylation sequons)). Functional subunits and domains are indicated and coloured as defined in the schematic representation as a function of the sequence number indicated above each functional unit. The position of S1/S2 and S2' cleavage sites are indicated. UH, upstream helix; CR, connecting region; iFP, internal fusion peptide; HR, heptad repeat; CH, central helix; BH, β-hairpin; SD3, SD3 subdomain; ST, stem helix; TM, transmembrane domain; CT, cytoplasmic tail. **(C)** Overall structure of the BwCoV protomer. Side views of the BwCoV protomer are shown in three directions. The different segments are colour coded consistent with panel B (S1⁰, tan; S1ᴬ, teal; S1ᴮ, dark salmon; S1ᶜ, yellow; S1ᴰ, sky blue; UH, light blue; CR, lilac; iFP, red; HR1, green; CH, purple; BH, light pink; SD3, gray; ST, gold).

CeCoV spike quaternary architecture shares certain features with alphacoronavirus spikes while being distinct from the spike of IBV (Figs 2c and D in S1 File).

Comparative structural analysis of individual domains of CeCoV and IBV revealed significant differences. CeCoV S1ᴬ shares the beta-sandwich core subdomain (S1ᴬ¹) typical of coronavirus S1ᴬ domains (Fig 2d). However, compared to IBV, CeCoV exhibits notable expansions in two S1ᴬ regions: residues 313–337 and 348–401 (based on BwCoV spike residue numbering). The latter of these regions forms a mini beta-sheet (secured to the other expanded region through a novel disulphide bond between Cys³²⁵ and Cys³⁹³) that extends to shape the top of the CeCoV spike trimer. Consequently, CeCoV S1ᴬ is ~70–90 residues larger than its IBV counterpart. While the structural relationship between CeCoV and IBV S1ᴬ domains is apparent, this is not the case for domain S1ᴮ. FoldSeek analysis [14] of CeCoV S1ᴮ against the PDB database reveals a closer structural relationship with the S1ᴮ domains of the alphacoronaviruses than to IBV (Fig 2c). The structural relatedness of CeCoV spikes to alpha- and deltacoronavirus S1ᴮ domains is also supported by FoldTree analysis [15] performed on all currently available experimentally characterized coronavirus S1ᴮ structures (Fig E in S1 File).

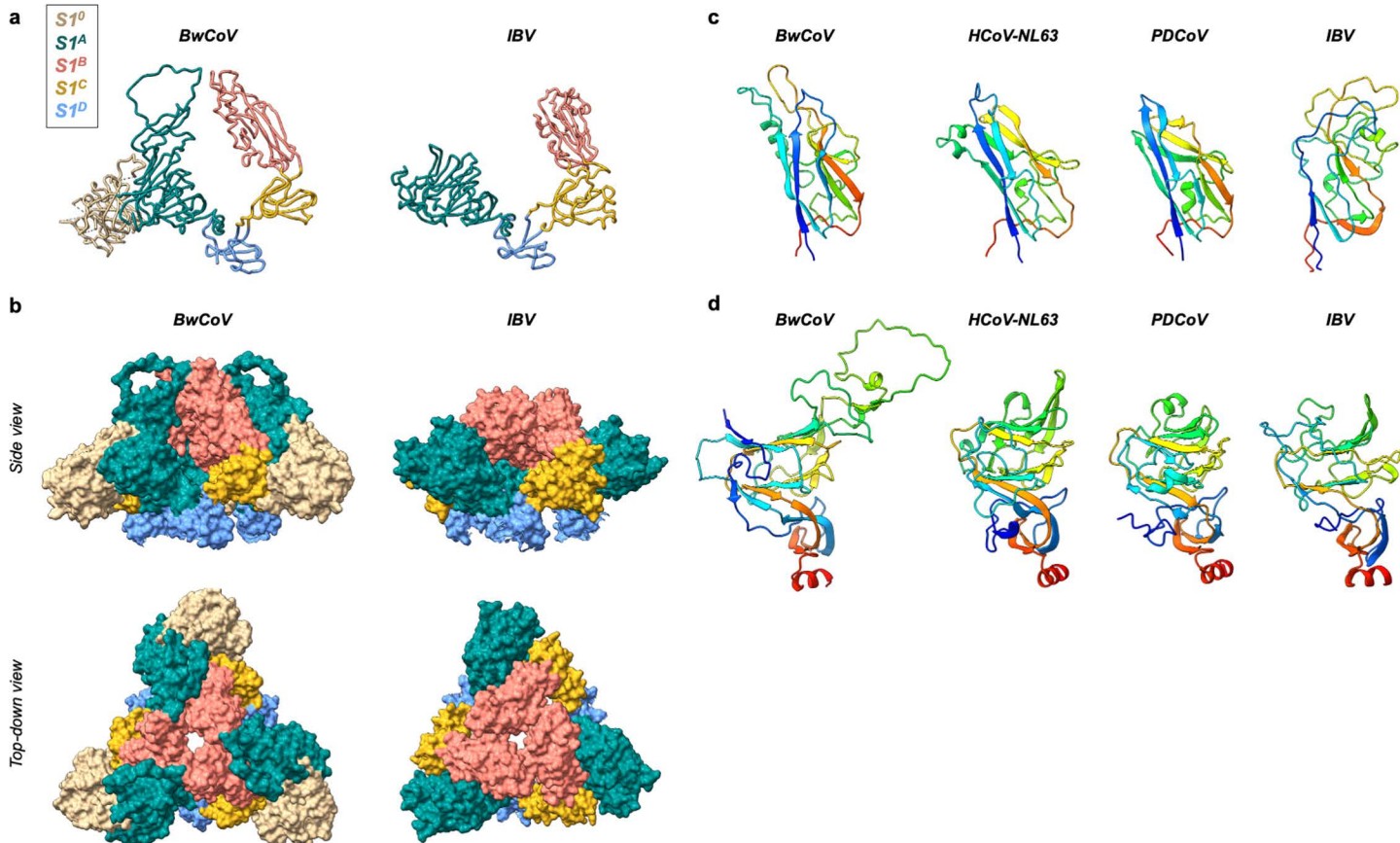

**Fig 2. CeCoV S1ᴬ and S1ᴮ domains are evolutionary distinct from IBV and adopt an alternative quaternary packing. (A)** Liquorice representation of the BwCoV and IBV S1 subunits. BwCoV S1ᴬ and S1ᴮ are tilted towards each other and interact through extended loops from both domains. BwCoV S1 has acquired an N-terminal extension (S1⁰). The individual domains of the S1 subunit are colour-coded: S1⁰, tan; S1ᴬ, teal; S1ᴮ, pink; S1ᶜ, golden; S1ᴰ, blue. **(B)** Side and top-down views of a surface representation of the S1 subunit of BwCoV and IBV within its trimeric context. S1ᴬ is intercalated by two S1ᴮ domains in BwCoV spike, whereas IBV S1ᴮ is predominantly exposed to solvent. Domains are coloured as in panel **A. (C)** Rainbow coloured cartoon representation of the aligned S1ᴮ domains of the gammacoronaviruses CeCoV (BwCoV) and IBV, the alphacoronavirus HCoV-NL63 and the deltacoronavirus PDCoV. Surprisingly, the CeCoV structure displays more resemblance to alpha- and deltacoronavirus S1ᴮ domains than to that of the gammacoronavirus IBV. **(D)** Rainbow coloured cartoon representation of the aligned S1ᴬ domains of the gammacoronaviruses CeCoV (BwCoV) and IBV, the alphacoronavirus HCoV-NL63 and the deltacoronavirus PDCoV. S1ᴬ domains across coronavirus genera display a similar core fold of this domain, as opposed to domain B shown in panel **c.**

### CeCoV S proteins exhibit a novel, host-derived domain tethered to domain S1ᴬ

Our cryo-EM reconstructions reveal that the CeCoV S1⁰ domain displays a previously unseen fold for CoV spike proteins by exhibiting a beta-barrel fold interspersed with loops and short helical elements. The sequence of this ~200 residue extension lacks homology to other known proteins, causing structure prediction programmes such as AlphaFold 3 [16] to output low-confidence prediction models for this domain (pLDDT scores <70 or <50) (Fig F in S1 File). The S1⁰ core consists of a beta-sandwich comprising a three- and four-stranded beta-sheet, with a third, skewed four-stranded beta-sheet positioned perpendicular to the beta-sandwich (Fig 3a and 3b). A loop that covers part of the S1ᴬ surface protrudes from this latter beta-sheet. The N-terminus of the protein packs against the membrane-proximal surface of this domain. As the most peripheral domain within the trimer, S1⁰ is the least well resolved region in both reconstructions (Fig C panel d in S1 File). Neither hetero nor local refinement appreciably improved the S1⁰ density, presumably due to the presence

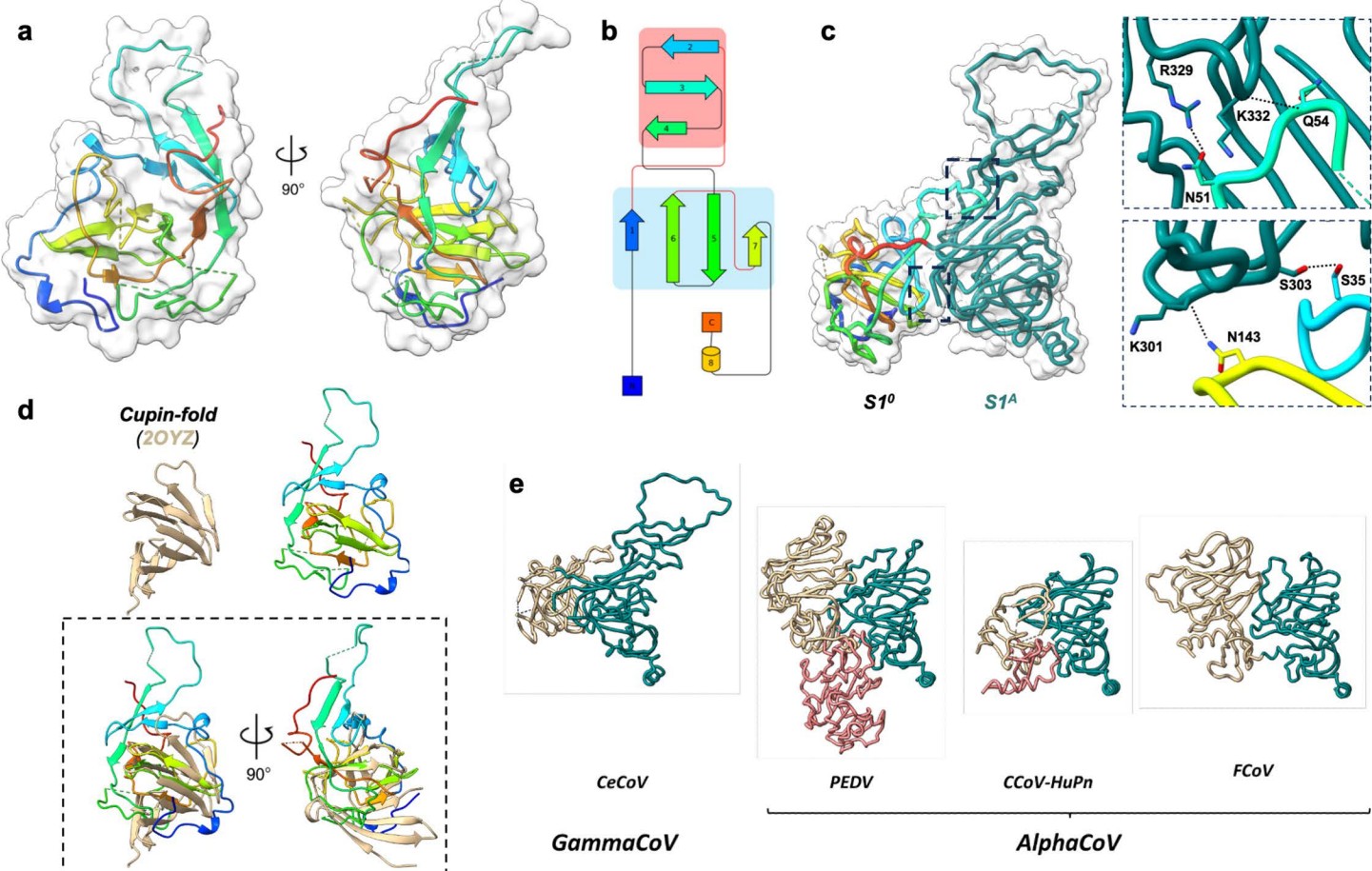

**Fig 3. The N-terminally appended S1⁰ domain is structurally homologous to cupin proteins and is tethered to domain S1ᴬ. (A)** Rainbow coloured cartoon representation of the N-terminal extension (S1⁰) of the BwCoV spike protein. Protein surface is shown as a white volume. Two orthogonal views are shown. Regions that could not be modelled are indicated as dashed lines. N-glycans are left out for clarity. **(B)** Topology plot of the BwCoV S1⁰ domain coloured as under panel **A. (C)** Side view of a liquorice representation of BwCoV S1⁰ (rainbow coloured) and S1ᴬ (teal), with zoom-in panels of the two regions of observed interdomain interactions. Zoomed-in areas are indicated on overall structure as two black boxes with dashed lines. **(D)** Cartoon representation of BwCoV S1⁰ and its top FoldSeek hit (PDB: 2OYZ). The protein domains are superimposed in the lower panel (dashed black box) and shown as two orthogonal views. **(E)** Side-by-side comparison of reported alphacoronavirus and CeCoV S1⁰ states. Structures are aligned on domain S1⁰. S1⁰ proximal state, tan; S1⁰ distal state, red; S1ᴬ, teal.

of disordered loops and multiple (N- and O-) glycosylation sites on the surface of CeCoV S1⁰ (*described in detail later on*) which resulted in certain S1⁰ regions being unmodelled in the atomic models (see Table B in S1 File for a complete overview, including unresolved S1ᴬ regions). At the same time, because we did not detect conformational heterogeneity across particles, the S1⁰ domain appears rigid relative to the trimer as a whole and is positioned against the S1ᴬ domain through a modest network of hydrogen bonds (Fig 3c).

Structural homology analysis using FoldSeek [14] and DALI [17] consistently aligns both S1⁰ domains with members of the cupin protein superfamily, known for broad functional diversity across all kingdoms of life, ranging from carbohydrate modification to gene regulation [18,19]. A FoldSeek search against the BwCoV S1⁰ model (residues 28–229) identifies the best hit within the experimental PDB database as protein VPA0057 from the marine bacterium *V. parahaemolyticus* (prob. 0.80 with 8.5% sequence identity; PDB: 2OYZ) (Fig 3d), while even stronger matches are found in the AlphaFold 2

prediction database (a cupin-2 domain-containing protein of the *K. pneumoniae* bacterium, prob. 0.87 with 9.1% sequence identity; AlphaFill model: AF-A0A0H3GUW4-F1-model_v4). The top FoldSeek hits are predominantly of bacterial origin (Figshare Data 1–2), an observation that is difficult to reconcile with the mammalian and avian host range of coronaviruses. At present, the evolutionary route by which this protein sequence entered a coronavirus genome therefore remains unclear.

## CeCoV S displays distinctive elements within the conserved S2 fusion machinery

The S2 fusion machinery of CeCoV S retains canonical features of the coronavirus fusion machinery with an upstream helix, an internal fusion peptide [20], the HR1 and HR2 regions, a central helix and a membrane proximal beta-hairpin (Fig 1c). Compared to the spike protein of IBV (strain M41), CeCoV spike is missing an S1/S2 furin cleavage site and the S2 subunit contains an insertion in the region directly upstream of the S2' site (BwCoV, Ser961 – Asp965; BdCoV, Ser984 – Gly988) (Fig G panel a in S1 File). Furthermore, relative to other gammaCoV spike proteins, an insertion of one heptad repeat occurred in HR1 (BwCoV, Thr1090 – Val1096; BdCoV, Thr1113 – Val1119) and three heptad repeats in HR2 (BwCoV, Gly1266 – Phe1286; BdCoV,Gly1289 – Phe1309), that jointly fold into the antiparallel helical bundle during fusion (Fig G panel b in S1 File). The latter 21-residue insertion forms a loop with a short $3_{10}$ helix that interacts with the membrane proximal part of the HR1 region (Fig G panel c in S1 File). Relative to IBV S, the CeCoV stem-proximal region also includes a four-residue deletion (IBV Pro958 - Ser961) which shortens one of the loops that faces the membrane. Predictably, HR2 and the region around the S1/S2 cleavage site remain unresolved in our cryo-EM reconstructions.

## CeCoV S proteins are extensively decorated with N-glycans

CeCoV spike sequences carry 37 (BwCoV) and 35 (BdCoV) predicted N-linked glycan sequons, accounting for roughly one-third of their molecular weights (Table C in S1 File). To gain an in-depth understanding of the N-linked glycosylation patterns, we performed mass spectrometry-based glycoproteomic analysis of recombinant soluble BwCoV and BdCoV S ectodomains to characterize their site-specific N-linked glycosylation patterns. We used a multi-protease digestion approach to create glycopeptides containing single N-glycan sites, detecting and identifying 33/37 N-linked glycan sites for BwCoV S and 32/35 for BdCoV S (Figs 4a and H panel a in S1 File). Glycan densities were visible in the EM reconstructions for 26/37 N-glycans on BwCoV and 29/35 on BdCoV S (Fig 4b), with missing densities typically mapping to unresolved regions of the maps. As expected, glycans are predominantly present on the S1 subunit, with fewer but highly conserved glycans detected on the surface of the S2 fusion subunit. A full overview of identified N-glycosylation sites is provided in Table D in S1 File and S1-S3 Data.

The glycoproteomics analyses show that there is extensive heterogeneity in glycoforms on CeCoV spikes ranging from unglycosylated asparagines and underprocessed oligo-mannose type glycans, to abundant, highly processed complex type glycosylation (Fig 4a). Notably, both $S1^0$ domains (BwCoV residues 24–229 and BdCoV residues 23–235) are decorated with clusters of underprocessed oligomannose type glycans, likely resulting from steric hindrance limiting access to Golgi-resident glycan processing enzymes [21,22]. Oligomannose type glycans are also found in other positions throughout the spike proteins where they may have arisen from protein-directed inhibition of glycan processing (*e.g.,* glycans attached to BwCoV residues Asn455 and Asn1209, and BdCoV residues Asn1208 and Asn1314). EM density often permitted modelling of glycan chains beyond the first core N-acetyl glucosamine (GlcNAc) (Fig H panel b in S1 File). In several cases, density for a fucose moiety connected to the first GlcNAc of a complex glycan was observed in the map and consequently included in the atomic model.

## CeCoV S1<sup>A</sup> domains present O-linked glycans on a mucin-like arch at the spike apex

Visualization of a Gaussian filtered EM map combined with the protein model illustrates the presence of an extensive CeCoV S glycan shield, which is particularly pronounced around the $S1^0$ domain (Fig 4c). This glycan shield displays EM

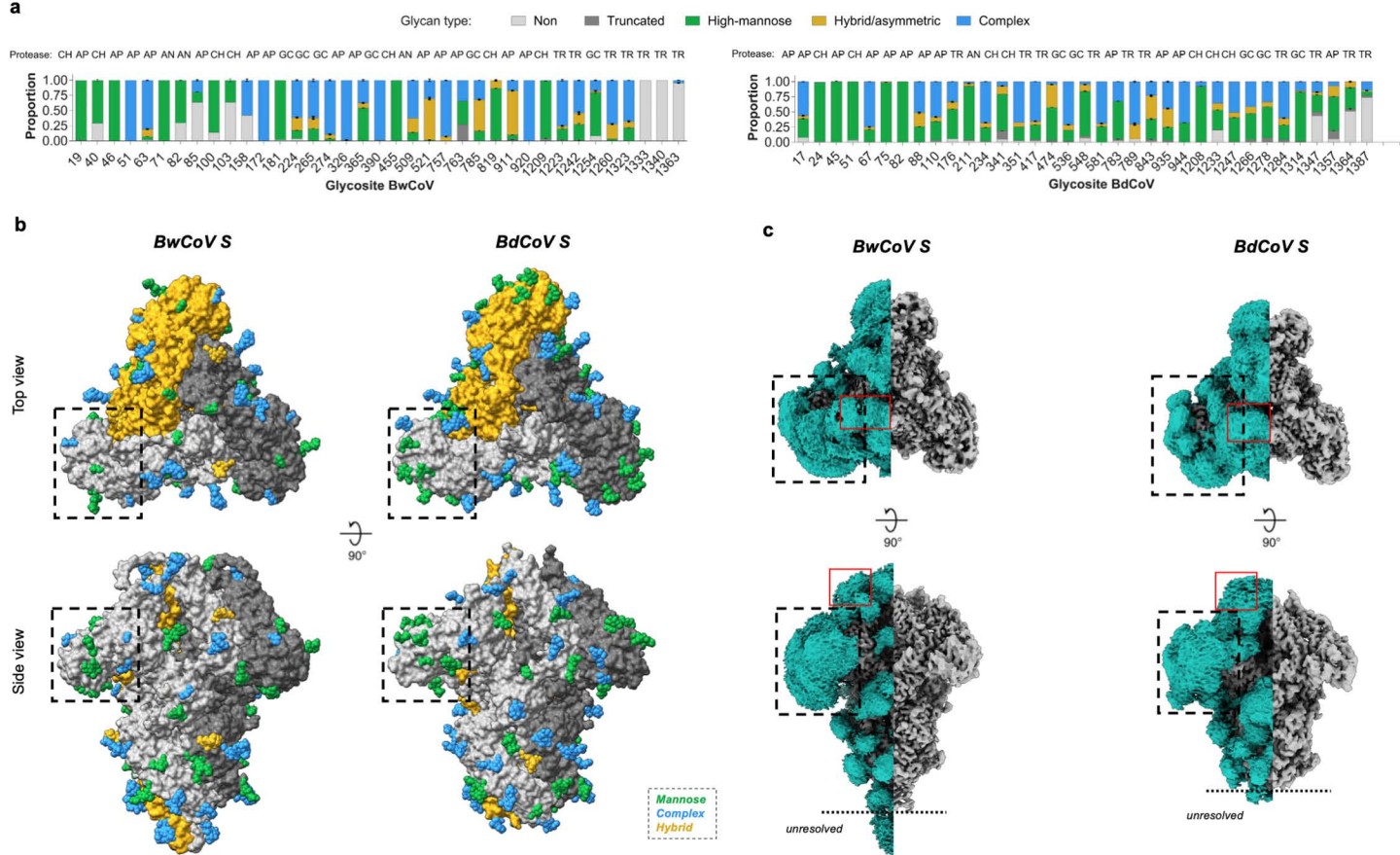

**Fig 4. CeCoV S proteins are highly N-glycosylated. (A)** Glycoproteomic analysis of N-linked glycosylation of BwCoV and BdCoV spike glycoprotein ectodomains. Panels display the relative proportion of detected glycoforms (non-glycosylated, white; high-mannose, green; hybrid/asymmetric, yellow; complex, blue) per glycosite. Error bars represent the standard deviation of the duplicate measurement. The selected protease dataset from which the data was extracted is indicated above the bars for each site: TR, trypsin; CH, chymotrypsin; AP, alpha-lytic protease; AN, AspN; GC, GluC. An overview of the data is available in Fig I panel b in S1 File, Data 1 and 2 at Figshare and S1 Data. **(B)** Surface representation of the trimeric CeCoV S reconstructions with modelled N-glycans displayed as spheres and colour-coded according to the most abundantly detected glycoform of each site as presented in panel **A.** Top-down and side views are shown. Protomers are coloured orange, grey and black. Zoomed-in panels highlight S1$^0$ domain surface which displays mainly high-mannose glycans. BwCoV S1$^0$ N-glycans 51, 63, 158 and 172, all present on the S1$^0$ surface that faces away from the viral membrane, do not present high mannose glycans. High mannose sugars that are resolved are positioned to the side of the S1$^0$ domain. **(C)** Visualization of a Gaussian filtered EM map segmented based on the atomic protein model (teal) and remaining EM density (grey) indicates the presence of an extensive CeCoV S glycan shield. Top and side views of the glycan shield of BwCoV (*left panel*) and BdCoV S (*right panel*). The unresolved region at the C-terminus of the protein is indicated. Black boxes indicate the position of the densely shielded domain S1$^0$, red boxes indicate the location of the mucin-like region.

density near the three-fold symmetry axis at the top of the spike protein (red boxes in Fig 4c) that does not correspond to any of the N-glycans we identified. Coincidentally, we noticed that corresponding to this area of the model, the CeCoV S proteins maintain a mucin-like region (MLR) within domain S1$^A$ that is not strictly conserved in sequence (Fig 5a and 5b). Such MLRs are characterized by a high frequency of Pro, Ser and Thr residues and are prone to O-glycosylation. As the resolution of the EM density corresponding to this region did not allow conclusive detection of such glycosylation, we performed mass spectrometry-based glycoproteomics to determine whether CeCoV S proteins contain O-linked glycosylation. O-linked glycosylation was detected with confidence in both CeCoV S samples at the MLR in S1$^A$ (Figs 5c and I panel a in S1 File), with a higher diversity of glycoforms observed for BdCoV (Fig I panels b and d in S1 File). The

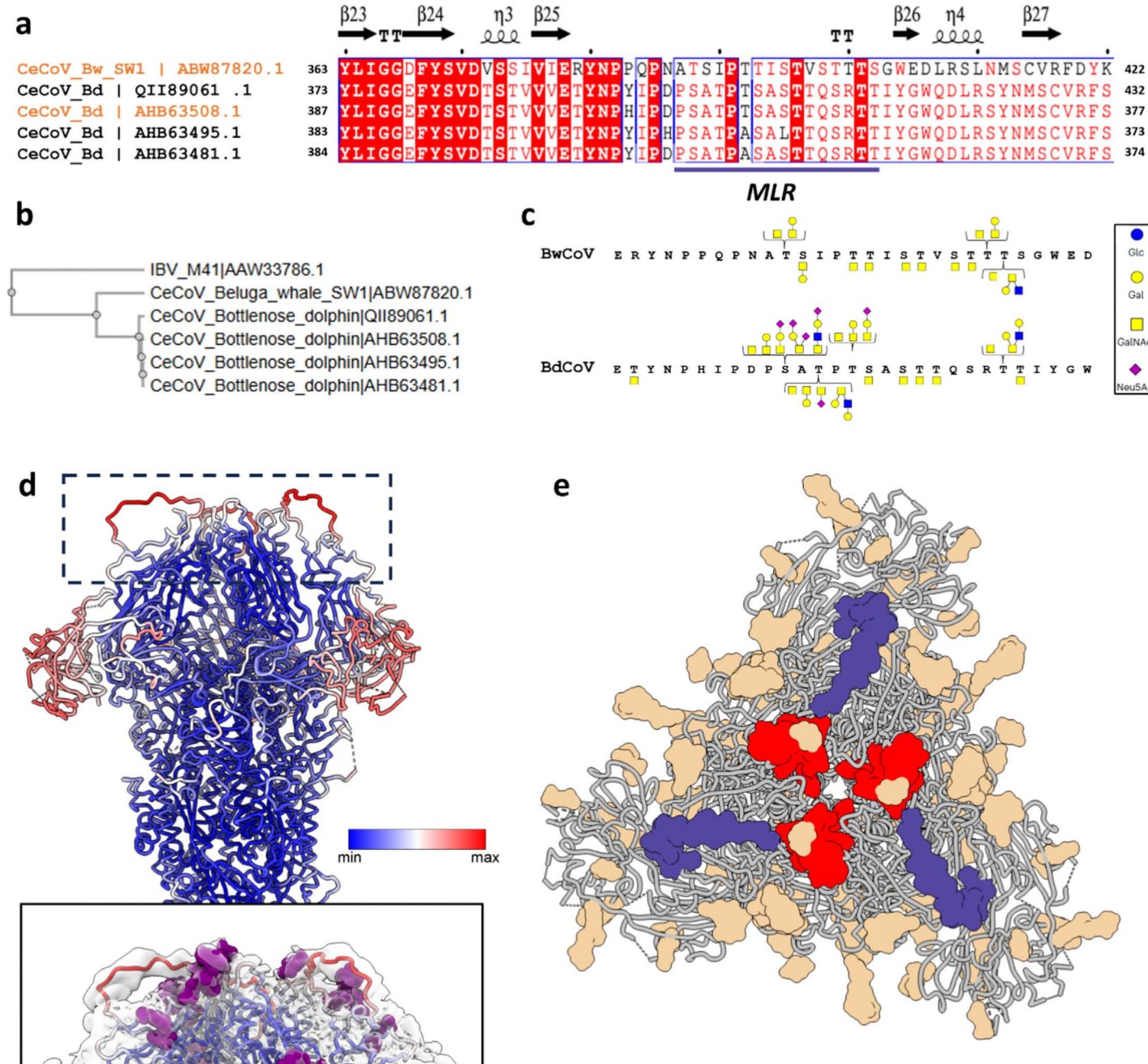

**Fig 5. CeCoV S S1ᴬ domains present O-linked glycans on a mucin-like arch at the spike apex. (A)** Sequence alignment of the five available CeCoV S sequences focusing on the mucin-like region (MLR) within domain S1ᴬ predicted to contain O-linked glycosylation (NetOGlyc 4.0 server[87]). Secondary structure assignment based on BwCoV S structure via ESPript 3.0[82]. Sequences were aligned with Clustal Omega[81] (EMBL-EBI). Protein sequences used to determine cryo-EM reconstructions of BwCoV and BdCoV S are marked in orange. **(B)** Phylogenetic tree of IBV spike protein (strain M41) and available CeCoV spike protein sequences. Sequences were aligned with Clustal Omega (EMBL-EBI[81]). **(C)** Overview of dominant O-linked gly-coforms within the MLR of BwCoV S1ᴬ (residues 361-385) and BdCoV S1ᴬ (residues 382-410). **(D)** *Top panel*: BwCoV spike protein model coloured by B-factor (minimal B-factor, blue; maximal B-factor) indicates flexibility of the MLR at the apex of the trimer and the periphery of domain S1⁰. *Lower panel*: side view of the apical face of BwCoV S, equivalent to the region indicated with the dashed black box in the top panel. The same liquorice representation as in the top panel and the N-glycans are indicated as purple spheres. The atomic model is fitted within a Gaussian filtered EM density and illustrates the relatively low quality of the map at this location. **(E)** Liquorice representation of BwCoV spike (grey) apex top-down view with different elements colour-coded (mucin-like arch, dark blue; putative receptor binding motif, red; N-glycan are coloured according to their glycoforms as shown in Fig 4).

prevalent O-linked glycoforms on residues of the MLR are schematically presented in Fig 5c. Interestingly, the O-glycosylated MLR is presented as a flexible arch at the apex of the trimeric spike assembly near the topologically conserved RBMs (Fig 5d, 5e). Besides the presence of O-glycosylation in the CeCoV mucin-like S1$^A$ loop, glycoproteomics analysis revealed O-glycans in various other positions of the CeCoV S ectodomains (Fig J panels a and b in S1 File). Comparison of these sites with the EM maps indicates density for many of these. The O-linked glycosylation complements the abundant N-glycosylation to create an extensive glycan shield for these mammalian gammacoronavirus spike proteins (see Fig J panel a in S1 File for an overview of experimentally confirmed N- and O-linked glycosylations of both CeCoV spike types). Remarkably, O-glycans were often detected at the Ser/Thr residue of the N-glycosylation sequons, further supporting an emerging theme in glycoproteomics that there is crosstalk between N- and O-linked glycosylation [23–25] (Fig J panel c in S1 File).

## Discussion

Coronavirus spike proteins have a common evolutionary origin and display structural differences that have arisen through extensive divergent evolution [26]. The conserved global spike protein architecture is characterized by an S1 crown that maintains the metastable S2 stalk in a pre-fusion conformation that is poised for membrane fusion upon receptor engagement and proteolytic cleavage [27,28]. Two types of pre-fusion spike assemblies can be distinguished: alpha- and deltacoronavirus spikes with intra-protomer S1 contacts, and beta- and gammacoronavirus spikes with inter-protomer S1 contacts [9,10,29,30]. Here, we describe the ectodomain structure of two cetacean coronavirus spike proteins that unexpectedly present an alternative quaternary assembly, in which elongated and outwardly displaced S1$^B$ domains are tilted upward in a manner analogous to, yet different from that of zoonotic betacoronavirus spikes [12,13]. Shielding of the topologically conserved RBM [11,31–40] on CeCoV S1$^B$ is achieved through contacts with extended protrusions emanating from the S1$^A$ beta-sandwich core of the same protomer, and positioning of a mucin-like S1$^A$ region at the trimer apex. Additionally, an N-terminal extension of cellular origin has displaced the S1$^A$ domain which now intercalates between neighbouring S1$^B$ domains. In doing so, CeCoV S has adopted an S1 protomer architecture reminiscent of that of alpha- and deltacoronavirus spike proteins but displays intriguing adaptations (Fig 2a).

Coinciding with this unusual quaternary assembly, CeCoV S has acquired an additional S1 crown domain with structural homology to cupin protein superfamily members. The cellular origin of S1$^0$ underscores the functional plasticity of coronavirus spikes. While the exact function of this domain remains unclear, this comprises the second known example of exaptation of a cellular gene by CoVs to function in concert with the S protein, following a previously observed structural relation between S1$^A$ and cellular galectins [41].

Spike proteins engage a diversity of receptors via a generally topologically conserved site on the S1$^B$ surface [11,31–34,36,42]. Structural comparison of CeCoV S1$^B$ indicates a structural relation to the equivalent domains of alpha- and deltacoronaviruses, whereas IBV S1$^B$ displays a distinct fold - suggesting that a recombination event separated these domains during evolution. With the notable exception of emerging zoonotic spikes for which dynamic S1$^B$ movement is observed [12,13], the receptor-binding site is inaccessible in trimeric pre-fusion spike assemblies, necessitating conformational changes prior to receptor engagement [9]. For human betacoronavirus HKU1 we recently discovered that exposure of the S1$^B$ RBM is triggered by allosteric engagement of the primary sialoglycan receptor by domain S1$^A$ [43]. CeCoV spike proteins appear to have evolved an even more intricate entry strategy in which S1$^A$ domains cover the RBMs of domain S1$^B$, while itself being tethered by the N-terminal S1$^0$ extension (Figs 2a, 2b and 5e). This contrasts observations made for the S1$^0$ domains of alphacoronaviruses, which are found to be more loosely attached to S1$^A$ (Figs 3e and H in S1 File) [44–46]. Notably, alphacoronaviral S1$^0$ domains stem from a gene duplication event of their S1$^A$ domain [9], making them evolutionary distinct to CeCoV S1$^0$. Whether or not CeCoV domain S1$^0$ can functionally replace S1$^A$ to serve as a trigger point for allosteric changes leading to exposure of the S1$^B$ RBM remains to be seen. These cryo-EM

reconstructions underline the structural plasticity of the coronavirus fusogen and hint at the possibility that modularity may stimulate exploration of alternative packing solutions of this metastable protein.

Glycosylation is an important and ubiquitous post-translational modification affecting protein folding, stability and function [47], and may guard viral protein surfaces from neutralizing antibodies elicited by the host immune system. The CeCoV S trimer is an abundantly glycosylated protein assembly with 35–37 N-glycosylated sequons per protomer. N-glycans are thought to mask immunogenic epitopes by covering them with host-derived glycotopes [9,22,48–50]. Similar to the N-terminal spike domains of most members from the alpha-, beta- and deltacoronavirus genera [9,44,48,49], the CeCoV S1$^0$ surface is the most exposed domain within the trimeric assembly and contains the highest degree of N-glycosylation. The local glycan abundance likely accounts for the observed level of underprocessed high-mannose glycans on this domain, similar to observations made for the S1$^A$ domain of MERS-CoV(49). While the paucity of CeCoV S sequences prohibits a comprehensive evolutionary analysis to identify regions under evolutionary pressure, antigenic mapping of other coronavirus spike proteins indicates that N-terminal domains are commonly targeted by neutralizing antibodies of the host immune system [51–56]. The presence of clustered oligomannose type N-glycans is reminiscent of that found for "evasion strong" viral glycoproteins of HIV-1, SIV and LASV [57–59]. While coronavirus S proteins generally do not present an effective glycan shield, CeCoV S proteins present high oligomannose abundance (Fig 4a), particularly within domain S1$^0$, and would appear more reminiscent of "evasion strong" viruses [49] (Table C in S1 File) based on their N-glycoforms alone.

We further detected several sites of O-glycosylation on CeCoV S (Fig J in S1 File), a feature largely absent in previously studied coronavirus spikes, with only trace levels observed in SARS-CoV-2 [22]. Both CeCoV S proteins present a mucin-like loop decorated with O-linked glycans at the apex of their trimers (Fig 5). Such regions have previously been reported for glycoproteins of filoviruses [60], flaviviruses [61], bunyaviruses [62], human pneumoviruses [63,64] and herpesviruses [65], where they contribute to pathogenicity [66]. In the case of CeCoV S, we speculate that these particularly situated O-glycans may shield the nearby receptor-binding motifs of S1$^B$ from neutralizing antibodies (Fig 5). Alternatively, the mucin-like region could play a role in initial attachment of the virus, which has been postulated for other viruses carrying mucin-like domains [67], but the exact role of CeCoV S O-glycosylation in viral infection remains to be experimentally explored.

Our results substantially expand our structural understanding of gammaCoV spike proteins and reveal that coronavirus spike proteins can adopt alternative quaternary assemblies and glycosylation patterns. The combination of modular domain acquisition and diverse glycosylation strategies enables functional adaptation and may facilitate host switching and immune evasion. These insights warrant further investigation into the receptor usage and spike entry mechanism of cetacean coronaviruses that may have allowed an ancestral avian virus to infect mammalian hosts in a manner reminiscent of PDCoV [68].

## Materials and methods

### Construct design, protein expression and purification

CeCoV spike sequences detected in a beluga whale (GenBank: ABW87820) and a bottlenose dolphin (GenBank: AHB63508) were human codon optimized, provided with 2P pre-fusion stabilising mutations [69] and ordered at Tsingke (Beijing, China). The BwCoV and BdCoV sequences were C-terminally truncated for soluble expression purposes (corresponding to BwCoV residues 24–1398 and BdCoV residues 23–1423) and ligated into a pCG2 expression vector with a CD5 signal peptide and C-terminally appended to a GCN4 trimerization motif and Strep-tag. For BdCoV, a second construct in which the C-terminus was truncated by an additional 53 residues was generated for structural purposes (BdCoV 23–1370).

Proteins were expressed in HEK 293F suspension cells by transient transfection with PEI (polyethyleneimine). The next day, cells were supplemented with 0.005% Primatone peptone and 2.25 mM Valproic acid. Media were harvested at

5 days post transfection, cleared from cell debris by centrifugation, supplemented with BioLock (IBA Life Sciences) and incubated overnight with Streptactin Sepharose beads (IBA Life Sciences) at 4°C. Beads were collected and washed with 0.1M Tris-HCl pH 8.0, 150 mM NaCl, 1 mM EDTA before elution and eluted with the same buffer supplemented with 2.5 mM D-biotin. Proteins were flash frozen and stored at -80°C prior to analysis.

## Negative stain and cryo-EM sample preparation and data collection

BwCoV and BdCoV SED-GCN4-ST proteins were diluted to 0.03 mg/ml for negative stain grid preparation. Three μl of the protein sample was applied to glow-discharged (15 mA, 20 sec, Pelco easiGlow) Carbon film 400 mesh Cu grids (Agar Scientific Ltd) and incubated for 30 sec. Grids were then washed twice with 10 ul of water, followed by 10 ul of uranyl formate and a final incubation with 10 ul of uranyl formate for 60 sec. Samples were imaged using a Talos L120C electron microscope operating at 120 keV. Spike protein ectodomains were concentrated to 1.3 mg/ml for cryo-EM grid preparation. Immediately before blotting and plunge freezing, 1 μl of 0.2% (w/v) fluorinated octyl maltoside (FOM) was added to 9 μl of each protein sample, resulting in a final FOM concentration of 0.02% (w/v). 3.5 μl of the protein-detergent mixture was then applied to glow-discharged Quantifoil R1.2/1.3 Cu 200 mesh grids (Quantifoil Micro Tools GmbH), blotted for 3.5 sec using 0° blot force at 100% humidity and plunge frozen into liquid ethane using Vitrobot Mark IV (Thermo Fisher Scientific). The data were collected on a Thermo Fisher Scientific Titan Krios G4 Cryo Transmission Electron Microscope (Cryo-TEM) equipped with a K3 Direct Electron Detector (Gatan, Inc.) operated at 300 keV.

In total, 8,500 and 2,400 movies were collected for BwCoV and BdCoV S, respectively, at a nominal magnification of ×105,000, corresponding to a calibrated pixel size of 0.836 Å/pix over a defocus range of −1.0 to −2.5 μm. To mitigate observed orientation bias of BdCoV SED, this data set was collected under a stage tilt of 33°. A full list of data collection parameters can be found in Table A in S1 File.

## Single particle image processing

Data processing was performed using the CryoSPARC Software package [70]. After patch-motion and CTF correction, particles were picked using a blob picker, extracted at 5x binning and subjected to 2D classification. Following 2D classification, particles belonging to class averages that displayed high-resolution detail were selected for ab-initio reconstruction into 4 classes. Particles belonging to the representative spike complex class were re-extracted at 1.3x binning. Spike complexes were subjected to non-uniform refinement with optimization of per-particle defocus and per-group CTF parameters (BwCoV S) or only per-group CTF parameters (BdCoV S). At this point, the global resolution of the spikes was 2.27 Å for BwCoV and 2.79 Å for BdCoV. Finally, reference-based motion correction was performed on the final particle stack of BdCoV S, improving the resolution with 0.14 Å to a final resolution of 2.65 Å for BdCoV S. For a more detailed processing methodology, see Figs B and C in S1 File.

## Model building and structure analysis

UCSF ChimeraX [71] (version 1.4) and Coot [72] (version 0.9.8.1) were used for model building. ModelAngelo [73] was used to generate initial spike model based on their sequence. The resulting model was then edited in Coot using the 'real space refine zone', 'regularize zone' tools and the 'carbohydrate module' [74]. To improve fitting, Namdinator [75] was utilized, using molecular dynamics flexible fitting of all models. Following this, iterative rounds of manual fitting in Coot and real space refinement in Phenix [76] were carried out to improve rotamer, bond angle and Ramachandran outliers. The Isolde plugin (version 1.4) [77] was used in UCSF Chimera to improve secondary structure assignment in domain S1⁰. During refinement with Phenix, secondary structure and non-crystallographic symmetry restraints were imposed. The final model was validated in Phenix with MolProbity [78,79] and fitted glycans were validated using Privateer [80,81]. Figures were made using UCSF ChimeraX(71). Structural homology searches were performed using the DALI [17] and FoldSeek servers [14]. Amino acid sequences were aligned using Clustal Omega [82] and illustrated with ESPRIPT 3.0 [83].

## N-glycosylated peptide preparation

Purified BwCoV and BdCoV proteins were digested with a panel of proteases according to the previously published protocol [84]. Briefly, 3.5 µg of protein per protease were denatured, reduced and alkylated in 1.5% SDC buffer (100 mM Tris pH 8.5, 1.5% sodium deoxycholate, 10 mM tris(2-carboxyethyl)phosphine, 40 mM iodoacetamide). Next, samples were diluted into 25 mM ammonium bicarbonate and digested with trypsin (Promega)+LysC (Sigma Aldrich) combination, chymotrypsin (Sigma Aldrich), alpha-lytic protease (Sigma Aldrich), GluC (Sigma Aldrich)+trypsin (Promega) combination or AspN (Roche) proteases at 1:50 protease: protein ratio at 37°C overnight. Peptides were desalted with 30 µm Oasis HLB 96-well plate (Waters). Desalted peptides were vacuum-dried and kept in -20°C until use.

## O-glycosylated peptide preparation

To obtain O-glycosylated peptides, BwCoV and BdCoV proteins were denatured, reduced and alkylated in 1.5% SDC buffer as described above and incubated for 18 hrs. at 37°C with 4 units/15 µg protein of PNGaseF (Roche) to remove all N-glycans. Subsequent digestion was performed with the same panel of proteases with the protocol described above.

## Mass spectrometry

Dried peptides were resuspended in 2% formic acid in water and 400 ng of peptides per protease sample were injected on either on Exploris 480 (ThermoFisher) or Orbitrap Fusion Lumos (ThermoFisher) mass spectrometer coupled to Ultimate3000 UHPLC system (ThermoFisher). The peptides were concentrated on an Acclaim Pepmap 100 C18 (5 mm × 0.3 mm, 5 µm, ThermoFisher Scientific) trap column and separated on in-house made 50-cm reverse-phase analytical column (75 µm inner diameter, ReproSil-Pur C18-AQ 2.4 µm resin [Dr. Maisch GmbH]) with a 90-min gradient starting at 4% buffer B to 15% B in 3 min, from 15% to 40% in 61 min, from 40% to 55% in 12 min, from 55% to 99% in 1 min, 99% washout in 4 min and re-equilibration back to 4% buffer B in 10 min where buffer A is 0.1% formic acid in water and buffer B is 0.1% formic acid in 80% acetonitrile (v/v). Exploris 480 parameters for the full MS scans were as follows: an AGC target of $5 \times 10^4$ at 120,000 resolution, scan range 350–2000 $m/z$, Orbitrap maximum injection time set to auto. The MS2 spectra were acquired for the ions in charge states 2+ to 8+, at a resolution of 30,000 with an AGC target of $5 \times 10^5$, maximum injection time set to auto, scan range was set from the first mass of 120 $m/z$, and dynamic exclusion of 14 s, precursor ion selection was at 1.4 $m/z$. The fragmentation was induced by stepped higher collision-induced dissociation with the NCE set to 10, 28, 45%. The parameters for Orbitrap Fusion Lumos were as described previously by Hurdiss et al., where EThcD fragmentation scheme was used. The data for the O-glycosylated peptides were acquired on Orbitrap Fusion Lumos system as described above but with the difference that oxonium ion product triggering was used prior to EThcD fragmentation scheme as described previously [85].

## N-glycoproteomic data analysis

The data were searched with Byonic (v.5.3.5) against a common 309 glycan database and BwCoV and BdCoV protein sequences. The protease specificity was set as follows: trypsin or trypsin+LysC- K/R with 2 missed cleavages; chymotrypsin – F/Y/W/M/L with 6 missed cleavages; alpha-lytic protease-T/A/S/V with 10 missed cleavages; AspN-D/E with 6 missed cleavages; trypsin+GluC-R/K/E/D with 6 missed cleavages. The search window was set to 10/20 ppm for MS1/MS2, respectively, and a False Discovery Rate (FDR) to 1%. Carbamidomethylation of cysteine was set as a fixed modification, methionine oxidation and glycan modifications were set as variable modifications allowing only one glycan and up to 3 variable modifications per peptide in total. Byonic glycopeptide hits were initially accepted if the Byonic score was ≥ 200, LogProb ≥3. Accepted glycopeptides were manually inspected for the quality of fragment assignments. The glycopeptide was considered true-positive if the appropriate b, y, c, and z fragment ions were matched in the spectrum, as well as the corresponding oxonium ions of the identified glycans. All glycopeptide identifications were merged into a

single non-redundant list per N-glycan site. Glycans were summed based on HexNAc content as truncated (up to maximum 2 HexNAc and 1 Fuc), high-mannose (2 HexNAc), hybrid/asymmetric (3 HexNAc), or complex (>3 HexNAc). Byonic results were used to build a spectral library in Skyline and extract peak areas for individual glycoforms from MS1 scans as described previously [84]. The semi-quantitative analysis of the glycosylation distribution was performed per site based on the peak area values. For the data presented in Fig 4a, protease dataset was selected based on overall signal for the corresponding N-glycan site. The acquired quantitative data were illustrated with GraphPad Prism 8 software.

## O-glycoproteomic data analysis

The data was searched with O-pair Search in MetaMorpheus (v1.0.6) with glycosearch option and Oglycan.gdb glycan database with 12 most common glycans selected. The search window was set to 10/20 ppm for MS1/MS2, respectively. Protease specificities were set as described above for trypsin and chymotrypsin samples; alpha-lytic protease, GluC and AspN samples were searched semi-specific with 24 missed cleavages allowed. Carbamidomethylation of cysteine was set as a fixed modification, methionine oxidation and deamidation of asparagine were set as variable modifications. The data type was set to HCD with child scan EThCD. Maximum allowed O-glycans per peptide were set to 7. Identified peptides were filtered on decoys and q-score (FDR1%). For the identification of O-glycans, only peptides with the O-glycan localization level 1 and level 1b were considered as identifications. The data were illustrated with GraphPad Prism 8 software.

## Analysis and visualization

Spike interface areas were calculated using PDBePISA [86]. The UCSF Chimera MatchMaker tool was used to obtain root mean square deviation values, using default settings. Domain rotations were calculated in UCSF ChimeraX. Figures were generated using UCSF ChimeraX [71]. Structural biology applications used in this project were compiled and configured by SBGrid [87].

## Supporting information

**S1 File. Fig A. Purification of BwCoV and BdCoV SED-GCN4-ST 2P proteins and corresponding negative stain data.** a) Anti-strep Western blot and SDS-PAGE of CeCoV spike protein ectodomains used for cryoEM. b) Negative stain micrographs of BwCoV and BdCoV spike ectodomains used for structural characterization. Micrographs display monodisperse particles that are of the expected size (~10 x 20 nm). Size bar represents 100 nm. **Fig B. Cryo-EM data processing pipelines in cryoSPARC10 for BwCoV (left panel) and BdCoV S (right panel). Fig C. Cryo-EM data processing of the BwCoV (left) and BdCoV (right) spike ectodomains.** a) FSC plots of BwCoV and BdCoV spike ectodomain reconstructions. b) Representative reference-free 2D class averages. c) Tightly contoured EM density maps for the spike ectodomains coloured according to local resolution which was calculated in cryoSPARC10 (countour level BwCoV: 0.156; BdCoV: 0.0544). d) Loosely contoured EM density maps for the spike ectodomains coloured according to local resolution which was calculated in cryoSPARC10 (countour level BwCoV: 0.0931; BdCoV: 0.0206). Part of the flexible S1[A] loop decorated with O-glycans at the apex of the trimer can be observed at this contour level (dashed black boxes). e) Angular distribution plot calculated in cryoSPARC[10] for particle projections in the globally refined map. f) Examples of EM density + model to illustrate resolution of maps. Top panels: BwCoV, bottom panels BdCoV. Panels left to right: S1[A], S1[B], S1[C], S2. **Fig D. CeCoV S quaternary packing is reminiscent of alphaCoV spike proteins.** Surface representations are shown of representative spike proteins from all four genera (IBV (6cv0), PDCoV (6b7n), SARS-CoV-2 (6vxx), HCoV-NL63 (5szs). Protomers of the trimeric models are coloured in the same manner for every protein. Models are ordered according to their packing mode: spike proteins displaying primarily inter-protomer contacts (beta-, gammaCoV) on the left and spike proteins displaying primarily intra-protomer contacts (alpha-, deltaCoV) on the right. The BwCoV spike structure is highlighted by a dashed red box. **Fig E. FoldTree[11] analysis of coronavirus S1[B] domains supports ancestral relation**

a) FoldTree analysis of experimentally characterized coronavirus S1$^B$ domains. b) Cartoon representation of aligned and rainbow-coloured atomic models of experimentally characterized coronavirus S1$^B$ domains. The models are shown in the order of the phylogeny for comparison. **Fig F. AlphaFold 3 structure prediction models of the BdCoV and BwCoV S1$^0$ domains does not reflect experimentally determined structures.** a) Cartoon representation of the experimentally determined structures of BdCoV and BwCoV S1$^0$, shown in the same (aligned) view. Unresolved regions are indicated by dashed lines, no glycans are shown for clarity. b) Cartoon representation of the top AlphaFold 3[14] models predicted for the amino acid sequences of domain S10 of the BdCoV and BwCoV spike proteins. Models were aligned to the experimentally determined structures shown in panel a and are coloured by pLDDT scores. The colour legend indicates the pLDDT confidence level (a higher score signifies a higher confidence). **Fig G. CeCoV S displays distinctive elements in conserved S2 fusion machinery.** a) Cartoon representation of the S2 stalk region of BwCoV (left panel) and IBV S (right panel) with the different S2 elements coloured as in Fig 1c for reference. The 5-residue insertion directly upstream of S2' site found in CeCoV S proteins (BwCoV residues Ser961 – Asp965; BdCoV residues Ser984 – Gly988) is shown in as red sticks coloured by heteroatom. b) The insertion of one heptad repeat in HR1 (BwCoV residues Thr1090 – Val1096; BdCoV residues Thr1113 – Val1119) is shown as red sticks coloured by heteroatom. c) The 21-residue insertion in HR2 (BwCoV residues Gly1266 – Phe1286; BdCoV residues Gly1289 – Phe1309) is shown as red sticks coloured by heteroatom.**Fig H. Detection of N-glycans on the surface of CeCoV spike glycoprotein.** a) Representative MS/MS spectra of the Byonic N-glycan identifications for BwCoV (top two panels) and BdCoV (lower two panels). b) EM density (grey mesh) zoned around the indicated modelled N-glycans of the BwCoV (left panels) and BdCoV (right panels) spike glycoproteins. **Fig I. Detection of O-glycans on the surface of the CeCoV spike glycoprotein.** a) Representative MS/MS spectra supporting Opair identifications of O-glycosylation sites. Top spectra show HCD fragmentation (b- and y-ions; blue and red, respectively) and identification of oxonium ions (light blue); bottom spectra show EThcD fragmentation where additionally c- and z-ions can be seen (yellow and orange, respectively). b) and d) Number of identified O-glycosites with the number of identified O-glycoforms per site for BwCoV and BdCoV S, respectively. Most identified O-glycosites carry only one glycoforms per site. c) and e) Number of O-glycans identified on the same peptide. **Fig J. Additional glycosylation data for CeCoV S proteins.** a) Overview of experimentally detected N- and O-glycosylation patterns of both CeCoV S proteins. N-glycan sites are depicted in green and O-glycan sites in purple. b) Overview of dominant O-linked glycoforms within BdCoV residues 379–410. c) Number of O-glycosylation sites identified within an N-glycan motif for both CeCoV S proteins. **Table A.** Cryo-EM data collection, refinement and validation statistics for global refinements. **Table B.** Overview of unresolved regions in CeCoV N-terminal domains. Topologically equivalent regions of the two spike proteins are listed within the same row. **Table C.** Overview of experimentally detected N-glycan abundance on coronavirus spike proteins. N-glycosylation calculated as 2.5 kDa addition to MW of glycoprotein. **Table D.** Overview and conservation of predicted N-linked glycosites across CeCoV BdCoV and BwCoV S proteins along with their corresponding experimental data.

(PDF)

**S1 Data. BwCoV_AUC_per_glycosite_all_proteases.**
(XLSX)

**S2 Data. BdCoV_AUC_per_glycosite_all_proteases.**
(XLSX)

**S3 Data. N_glycol_site_detection_per_protease.**
(XLSX)

**S4 Data. O_glycans_Opair_pooledData.**
(XLSX)

## Acknowledgments

We would like to thank the staff at the Utrecht University Electron Microscopy Centre and Willem Noteborn at Netherlands Centre for Electron Nanoscopy (NeCEN) for technical support with EM data collections.

## Author contributions

**Conceptualization:** Ruben J.G. Hulswit, Berend Jan Bosch, Daniel L Hurdiss.

**Data curation:** Ruben J.G. Hulswit, Tatiana M. Shamorkina.

**Formal analysis:** Ruben J.G. Hulswit, Tatiana M. Shamorkina.

**Funding acquisition:** Ruben J.G. Hulswit, Tatiana M. Shamorkina, Frank J.M. van Kuppeveld, Joost Snijder, Berend Jan Bosch, Daniel L Hurdiss.

**Investigation:** Ruben J.G. Hulswit, Tatiana M. Shamorkina, Joline van der Lee, Floor Rosman, Lisanne S Wetzels, Daniel L Hurdiss.

**Methodology:** Ruben J.G. Hulswit, Tatiana M. Shamorkina, Joost Snijder, Daniel L Hurdiss.

**Project administration:** Ruben J.G. Hulswit.

**Resources:** Frank J.M. van Kuppeveld, Joost Snijder, Berend Jan Bosch, Daniel L Hurdiss.

**Software:** Ruben J.G. Hulswit, Daniel L Hurdiss.

**Supervision:** Ruben J.G. Hulswit, Joost Snijder, Berend Jan Bosch, Daniel L Hurdiss.

**Validation:** Ruben J.G. Hulswit, Tatiana M. Shamorkina, Joline van der Lee, Floor Rosman, Lisanne S Wetzels.

**Visualization:** Ruben J.G. Hulswit, Tatiana M. Shamorkina, Daniel L Hurdiss.

**Writing – original draft:** Ruben J.G. Hulswit, Berend Jan Bosch, Daniel L Hurdiss.

**Writing – review & editing:** Ruben J.G. Hulswit, Tatiana M. Shamorkina, Joost Snijder, Berend Jan Bosch, Daniel L Hurdiss.

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
