## [Decision Letter · Decision Letter 0]

18 Nov 2025

PPATHOGENS-D-25-02465

Cetacean coronavirus spikes highlight S glycoprotein structural plasticity

PLOS Pathogens

Dear Dr. Hulswit,

Thank you for submitting your manuscript to PLOS Pathogens. We are pleased to inform you that the reviewers provided positive comments and are both in favor of publications with minor revisions only. We therefore invite you to submit a revised version of the manuscript that addresses the points raised during the review process.

We look forward to receiving your revised manuscript.

Kind regards,

Félix A. Rey

Academic Editor

PLOS Pathogens

Ronald Swanstrom

Section Editor

PLOS Pathogens

Sumita Bhaduri-McIntosh

Editor-in-Chief

PLOS Pathogens

orcid.org/0000-0003-2946-9497

Michael Malim

Editor-in-Chief

PLOS Pathogens

orcid.org/0000-0002-7699-2064

**Journal Requirements:**

At this stage, the following Authors/Authors require contributions: Ruben J.G. Hulswit, Tatiana Shamorkina, Joline van der Lee, Floor Rosman, Lisanne S Wetzels, Frank J.M. van Kuppeveld, Joost Snijder, Berend Jan Bosch, and Daniel L Hurdiss. Please ensure that the full contributions of each author are acknowledged in the "Add/Edit/Remove Authors" section of our submission form.

4) We notice that your supplementary Figures, and Tables are included in the manuscript file. Please remove them and upload them with the file type 'Supporting Information'. Please ensure that each Supporting Information file has a legend listed in the manuscript after the references list.

5) In the online submission form, you indicated that All reagents and relevant data are available from the authors upon request. All PLOS journals now require all data underlying the findings described in their manuscript to be freely available to other researchers, either

1. In a public repository

2. Within the manuscript itself

3. Uploaded as supplementary information.

2) If any authors received a salary from any of your funders, please state which authors and which funders..

7) Please send a completed 'Competing Interests' statement, including any COIs declared by your co-authors. If you have no competing interests to declare, please state "The authors have declared that no competing interests exist". Otherwise please declare all competing interests beginning with the statement "I have read the journal's policy and the authors of this manuscript have the following competing interests"

**Reviewers' Comments:**

Reviewer's Responses to Questions

**Part I - Summary**

Reviewer #1: In this manuscript, Hulswit et al. report high-resolution cryo-EM structures of two cetacean coronavirus spike proteins, showing that these spike trimers adopt a unique quaternary architecture, different from that of other coronaviruses. They have upright oriented S1B domains covered by S1A domains, and also a novel N-terminal S10 domain that resembles a cupin fold. Glycoproteomic analysis has identified a mucin-like loop at the trimer apex with extensive N-glycosylation and O-glycosylation contributing to the dense glycan shield that map help evade host immune responses. The authors suggest that these interesting structural features offer new insights into the evolutionary mechanisms for coronaviruses to expand their host range.

The manuscript is very well written and all the experiments are relatively straightforward and technically sound. These new structures indeed help advance our understanding of structure-function of diverse coronavirus spike proteins.

Reviewer #2: In this study, the authors determined high‑resolution cryo‑EM structures of spike glycoproteins from two cetacean gammacoronaviruses that infect marine mammals (beluga whale and bottlenose dolphin). Their structural analyses revealed an alternative quaternary assembly in which the S1B domains are tilted upright but shielded by extended loops from the S1A domain and a mucin-like apex, effectively concealing the conserved receptor-binding motifs. They further identified a novel ~200‑residue N‑terminal domain (S10) of host origin, structurally related to cupin proteins. Complementary glycoproteomic profiling showed that these spikes are heavily decorated with N‑glycans (>100 per trimer) and O‑glycans within the mucin‑like loop, forming a dense glycan shield. Together, these results demonstrate that cetacean coronaviruses have evolved unique structural and glycosylation features that expand the known diversity of coronavirus spikes, providing new insights into how these viruses adapt to novel hosts.

This is a well-written and straightforward manuscript that describes new insights into the diversity of coronavirus spike protein structure and architecture. The structural studies are performed to a high standard, and the analyses are thorough and thoughtful. Below are some minor comments and suggestions that may improve the manuscript.

**Part II – Major Issues: Key Experiments Required for Acceptance**

Please use this section to detail the key new experiments or modifications of existing experiments that should be absolutely required to validate study conclusions.required to validate study conclusions.

Reviewer #1: (No Response)

Reviewer #2: No major issues

**Part III – Minor Issues: Editorial and Data Presentation Modifications**

Reviewer #1: There are only a few minor issues that should be addressed.

1. Discussion on the similarity between the S10 domain and cupin fold is not very convincing, as they do not look like having a same fold, at least, as shown in Fig.3d (perhaps a better view would help?). The best matches are of bacterial origin and show only 8-9% sequence identity (not sure whether there are any conserved motifs critical for the cupin fold). So trying to conclude that “the S10 sequence originates from either a mammalian or avian host given the observed species tropism of coronaviruses” sounds a bit stretchy.

2. In the figure legend –“Fig 1 SPA structure determination of BwCoV SED at 2.27 Å and BdCoV SED at 2.65 Å”, it seems to be the only place where SPA is mentioned and it should be spelt out (single particle analysis?) for general readers.

3. The sentences –“Neither hetero or local refinement appreciably improved the S10 density, presumably due to intrinsic local protein flexibility and abundance of (N- and O-) glycosylation sites on the surface of CeCoV S10 (described in detail later on) causing several S10 regions to be absent in the final atomic models (see Table S2 for a complete overview including unresolved S1A regions). The lack of observed conformational heterogeneity of domain S10 relative to the rest of the spike protein complex underlines the existence of a singular S10 state, packed against the S1A domain by a modest network of hydrogen bonds involving main and side chain atoms (Fig. 3c)” are a bit confusing for general readers. First, the S10 density could not be improved because of protein flexibility, and then, “lack of observed conformational heterogeneity of domain S10”. What they meant is probably that S10 domain is rigid relative to the trimer, but it has some disordered loops.

Reviewer #2: 1. The clashscores for the structures are a little high, particularly for the higher-resolution BwCoV S model (21.00). The authors may try to loosen the geometric constraints during refinement to improve the score.

2. The figures are, in general, clear and nicely illustrate the key points being made in the text. However, I feel Fig 1a could be improved. At a minimum, the EM maps could be removed so that readers can focus on the architecture of the spike. Improved coloring or surface representation may also help.

3. line 24. The zero in S10 needs to be superscripted.

4. Something seems off in lines 257-258: “Glycan densities were visible in the EM reconstructions for 26/37 N-unresolved regions of the maps.”

5. Out of curiosity, were the 2P stabilising mutations necessary? Beneficial?

PLOS authors have the option to publish the peer review history of their article (what does this mean? ). If published, this will include your full peer review and any attached files.). If published, this will include your full peer review and any attached files.

**Do you want your identity to be public for this peer review?** For information about this choice, including consent withdrawal, please see our For information about this choice, including consent withdrawal, please see our Privacy Policy ..

Reviewer #1: No

Reviewer #2: No

**Figure resubmission:**
---

## [Editor Report · Decision Letter 1]

29 Dec 2025

Dear Dr. Hulswit,

We are pleased to inform you that your manuscript 'Cetacean coronavirus spikes highlight S glycoprotein structural plasticity' has been provisionally accepted for publication in PLOS Pathogens.

Best regards,

Félix A. Rey

Academic Editor

PLOS Pathogens

Ronald Swanstrom

Section Editor

PLOS Pathogens

Sumita Bhaduri-McIntosh

Editor-in-Chief

PLOS Pathogens

orcid.org/0000-0003-2946-9497

Michael Malim

Editor-in-Chief

PLOS Pathogens

orcid.org/0000-0002-7699-2064
---

## [Editor Report · Acceptance letter]

Dear Dr. Hulswit,

We are delighted to inform you that your manuscript, "Cetacean coronavirus spikes highlight S glycoprotein structural plasticity," has been formally accepted for publication in PLOS Pathogens.

Best regards,

Sumita Bhaduri-McIntosh

Editor-in-Chief

PLOS Pathogens

orcid.org/0000-0003-2946-9497

Michael Malim

Editor-in-Chief

PLOS Pathogens

orcid.org/0000-0002-7699-2064